# Pharmacological Inhibition of miR-130 Family Suppresses Bladder Tumor Growth by Targeting Various Oncogenic Pathways via PTPN1

**DOI:** 10.3390/ijms22094751

**Published:** 2021-04-29

**Authors:** Yuya Monoe, Kentaro Jingushi, Akitaka Kawase, Takayuki Hirono, Ryo Hirose, Yoshino Nakatsuji, Kaori Kitae, Yuko Ueda, Hiroaki Hase, Yuichi Abe, Jun Adachi, Takeshi Tomonaga, Kazutake Tsujikawa

**Affiliations:** 1Laboratory of Molecular and Cellular Physiology, Graduate School of Pharmaceutical Sciences, Osaka University, Osaka, 1-6 Yamadaoka, Suita, Osaka 565-0871, Japan; monoe-y@phs.osaka-u.ac.jp (Y.M.); kawase-a@phs.osaka-u.ac.jp (A.K.); hirono-t@phs.osaka-u.ac.jp (T.H.); hirose-r@phs.osaka-u.ac.jp (R.H.); nakatuji-y@phs.osaka-u.ac.jp (Y.N.); kitae@phs.osaka-u.ac.jp (K.K.); yukoueda@phs.osaka-u.ac.jp (Y.U.); hase-h@phs.osaka-u.ac.jp (H.H.); tujikawa@phs.osaka-u.ac.jp (K.T.); 2Laboratory of Proteome Research, National Institute of Biomedical Innovation, Health and Nutrition, Ibaraki, Osaka 567-0085, Japan; yuichi_abe@nibiohn.go.jp (Y.A.); jun_adachi@nibiohn.go.jp (J.A.); tomonaga@nibiohn.go.jp (T.T.); 3Laboratory of Proteomics for Drug Discovery, Center for Drug Design Research, National Institute of Biomedical Innovation, Health and Nutrition, Ibaraki, Osaka 567-0085, Japan

**Keywords:** miR-130 family, locked nucleic acid, bladder cancer, Src, PTPN1

## Abstract

Previously, we have revealed that the miR-130 family (miR-130b, miR-301a, and miR-301b) functions as an oncomiR in bladder cancer. The pharmacological inhibition of the miR-130 family molecules by the seed-targeting strategy with an 8-mer tiny locked nucleic acid (LNA) inhibits the growth, migration, and invasion of bladder cancer cells by repressing stress fiber formation. Here, we searched for a functionally advanced target sequence with LNA for the miR-130 family with low cytotoxicity and found LNA #9 (A(L)^i^i^A(L)^T(L)^T(L)^G(L)^5(L)^A(L)^5(L)^T(L)^G) as a candidate LNA. LNA #9 inhibited cell growth in vitro and in an in vivo orthotopic bladder cancer model. Proteome-wide tyrosine phosphorylation analysis suggested that the miR-130 family upregulates a wide range of receptor tyrosine kinases (RTKs) signaling via the expression of phosphorylated Src (pSrc^Tyr416^). SILAC-based proteome analysis and a luciferase assay identified protein tyrosine phosphatase non-receptor type 1 (PTPN1), which is implicated as a negative regulator of multiple signaling pathways downstream of RTKs as a target gene of the miR-130 family. The miR-130-targeted LNA increased and decreased PTPN1 and pSrc^Tyr416^ expressions, respectively. PTPN1 knockdown led to increased tumor properties (cell growth, invasion, and migration) and increased pSrc^Tyr416^ expression in bladder cancer cells, suggesting that the miR-130 family upregulates multiple RTK signaling by targeting PTPN1 and subsequent Src activation in bladder cancer. Thus, our newly designed miR-130 family targeting LNA could be a promising nucleic acid therapeutic agent for bladder cancer.

## 1. Introduction

Bladder cancer is the ninth most common malignant disease and the 13th most common cause of cancer-related deaths worldwide. Bladder cancer accounts for an estimated 500,000 new cases and 200,000 deaths worldwide [1]. Bladder cancer has considerable intratumoral heterogeneity at the genomic, transcriptional, and cellular levels, which is difficult to quantify [2,3,4]. Therefore, therapeutic targets for bladder cancer should include genes that regulate various cancer-related signaling pathways.

Over the past decade, microRNAs (miRNAs) have been reported to present altered expression patterns in malignant tumor cells. It is now considered that miRNA dysregulation influences critical molecular pathways involved in tumor progression and metastasis in a wide range of cancer types. Therefore, altering miRNA levels in cancer cells has promising potential as a therapeutic strategy for cancer [5]. Several studies have shown that the inhibition of miRNA family members through their common seed sequence significantly affects tumor progression [6,7]. The miR-130 family (miR-130b, miR-301a, and miR-301b) has also been shown to contribute to cancer progression, including that of bladder cancer [8,9,10]. Furthermore, we recently found that the pharmacological inhibition of the miR-130 family by a seed-targeting 8-mer locked nucleic acid (LNA) sequence suppressed the cell growth of bladder cancer [11]. Because of its tiny 8-mer seed-targeting LNA, target specification and off-target effects are of concern. Therefore, to obtain the functionally advanced miR-130 family targeted LNA, we designed several elongated LNA sequences with chemical modifications and examined the efficacy of pharmacological inhibition of the miR-130 family molecules.

We report here that the functionally advanced miR-130 family targeted LNA suppressed the phenotypes of bladder cancer cells both in vitro and in an orthotopic bladder cancer model in vivo, indicating that the miR-130 family targeted LNA is expected to be a promising therapeutic drug for bladder cancer.

## 2. Results

### 2.1. miR-130 Family Targeted LNA Suppresses Tumor Growth in an Orthotopic Bladder Cancer Model

To obtain miR-130 family targeted LNA with advanced binding activity, we designed 10 LNA-containing nucleotides that target the seed sequence of the miR-130 family and screened their activity using a luciferase assay in the bladder cancer cell line 5637 at a concentration of 50 nM. The luciferase activity derived from the miR-301a reporter vector co-transfected with a negative control LNA was significantly lower than that of the mock reporter vector alone. The luciferase activity of the miR-301a reporter vector was recovered by co-transfection with the miR-130 family targeted LNAs, including LNA #1, which was previously found to have an inhibitory effect on the function of miR-130 family molecules (Appendix A). Compared to the negative control, downregulation of Rennila luciferase activity showed cytotoxic effects. Among the 10 LNAs, LNA #1, #2, #7, #8, and #9 had low cytotoxic activity (Appendix A). Although LNA #1 had no significant effect on the growth of 5637 cells at 5 nM, LNA #9 suppressed the growth of 5637 cells (Appendix A). As LNA #9 showed a specific miR-301a-binding affinity with low cytotoxic activity and suppressed the proliferation of 5637 cells more strongly than LNA #1, we used LNA #9 (hereafter, referred to as miR-130F LNA) to investigate its anti-tumor potential in vitro and in vivo.

The anti-tumor potential of the miR-130F LNA was investigated in the bladder cancer cell lines 5637 and UM-UC-3. The miR-130F LNA successfully suppressed the proliferation of the cells of the two bladder cancer cell lines in a dose-dependent manner in vitro (Figure 1A,B). Since miR-130 family has been reported as oncomiR in colon cancer [12,13], we examined the effect of miR-130F on colon cancer cell lines HCT-116 and DLD-1 cells. The miR-130F LNA also suppressed the proliferation of colon cancer cell lines (Appendix A). Next, we examined the effect of miR-130F LNA in vivo using a subcutaneous transplant model of 5637 cells (Figure 1C). The tumor volumes and weights, which increased by subcutaneous administration of control LNA, were suppressed by the administration of miR-130F LNA (Figure 1D,E). Next, we evaluated the anti-tumor potential of miR-130F LNA using an orthotopic bladder cancer model of UM-UC-3 cells with stable luciferase gene expression (UM-UC-3 Fluc cells) (Figure 1F). The orthotopic bladder cancer model that we established responded well to transurethral administration of cisplatin, which was used as a positive control (Appendix A). The anti-tumor activity of miR-130F LNA was determined based on the decreased luciferase activity on day 14 by in vivo imaging. These results indicate that miR-130F LNA may be a novel nucleic acid therapeutic for bladder cancer.

### 2.2. miR-130 Family Upregulates Various Receptor Tyrosine Kinases in Bladder Cancer Cells

Previously, we showed that the miR-130 family functions as an oncomiR by targeting phosphatase and tensin homolog deleted from chromosome 10 (PTEN) and protein-tyrosine phosphatase non-receptor type 11 (PTPN11), which leads to upregulated migration and invasion activities in bladder cancer cells [8]. To fully understand the oncomiR function of the miR-130 family in bladder cancer, we focused on tyrosine-phosphorylated proteins, which govern major tumor-promoting pathways, such as epidermal growth factor receptor (EGFR) and vascular endothelial growth factor receptor (VEGFR) signaling. Proteome-wide tyrosine phosphorylation analysis was conducted using miRNA mimics of miR-301a, miR-301b, and miR-130b in UM-UC-2 cells. Fifty proteins showed upregulated tyrosine phosphorylation by miR-130 family mimic (Table 1, fold-change >1.2). Gene enrichment analysis (WikiPathways, https://www.wikipathways.org/index.php/WikiPathways, accessed on 8 March 2021) showed that the miR-130 family mimics affected a broad range of receptor tyrosine kinases, which constitute the major tumor-promoting pathway in cancer (Appendix A). Moreover, database analysis (http://xena.ucsc.edu, accessed on 8 March 2021) showed that miR-130 family expression was correlated with a broad range of phosphorylated receptor tyrosine kinase expression in bladder cancer specimens (Appendix A). Proto-oncogene tyrosine-protein kinase Src functions as the hub of a vast array of signaling pathways [14]. Although, the miR-130 family mimics had no significant effect on phosphorylation levels of epidermal growth factor receptor (EGFR^Tyr1068^, Appendix A), the miRNA mimic targeting the miR-130 family upregulated the phosphorylation levels of Src^Tyr416^ (Figure 2A, Table 1) and its downstream effector molecule Akt (Akt^Ser473^ and Akt^Thr308^) in UM-UC-2 cells (Figure 2B). On the other hand, miR-130F LNA inhibited the phosphorylation levels of Src and Akt in 5637 cells (Figure 2C), suggesting that the miR-130 family functions as an oncomiR by targeting various tumor-promoting pathway through Src phosphorylation, and miR-130F LNA could inhibit the downregulation of Src phosphorylation in bladder cancer cells.

### 2.3. miR-130 Family Targets PTPN1 in Bladder Cancer Cells

To elucidate the targets of the miR-130 family in bladder cancer, we conducted stable isotope labeling using amino acids in cell culture (SILAC)-based proteome analysis using negative control, miR-301a, miR-301b, or miR-130b mimic-transfected UM-UC-2 cells. The results of SILAC-based proteome analysis were further compared with the miRNA-target prediction database (http://www.targetscan.org/vert_72/, accessed on 28 January 2021). As shown in Figure 3A, 166, 79, and 137 proteins were downregulated by miR-130b, miR-301a, and miR-130b mimics, respectively. Of these, 41 proteins were shared among the three miR-130 family members (Table 2 and Appendix A).

We focused on PTPN1, implicated as a negative regulator of multiple signaling pathways downstream of receptor tyrosine kinases [15,16,17]. We then confirmed whether PTPN1 is a target molecule of the miR-130 family in a dual-luciferase reporter assay using a reporter vector containing the 3′-untranslated region (UTR) region of the PTPN1 gene (Figure 3B). The luciferase activity of the PTPN1 reporter vectors co-transfected with miR-130 family mimics was significantly lower than those co-transfected with the negative control mimic (Figure 3C). Western blot analyses showed that transfection with the miR-130 family mimic decreased PTPN1 expression in UM-UC-2 cells (Figure 3D). We then evaluated the effect of miR-130F LNA on PTPN1 protein expression. The luciferase activity of the PTPN1 reporter vectors co-transfected with the negative control LNA was significantly lower than that of the mock reporter vector alone. The activity was rescued by co-transfection with the miR-130F LNA (Figure 3E). Moreover, Western blot analyses showed that the transfection of miR-130F LNA increased PTPN1 protein expression in 5637 cells (Figure 3F). These data suggest that the miR-130 family targets PTPN1 and miR-130F LNA can restore PTPN1 expression downregulated by miR-130 family molecules.

### 2.4. PTPN1 Functions as a Tumor Suppressor in Bladder Cancer Cells

To examine the role of PTPN1 in bladder cancer cells, PTPN1 knockdown experiments were conducted. PTPN1 knockdown (Figure 4A) upregulated the growth (Figure 4B), invasion (Figure 4C), and migration activities (Figure 4D) of UM-UC-2 cells. Moreover, PTPN1 knockdown upregulated the phosphorylation levels of Src^Tyr416^ (Figure 4E), suggesting that the miR-130 family functions as a tumor promoter by targeting PTPN1 in bladder cancer cells.

## 3. Discussion

In the present study, we obtained a functionally advanced miR-130 family targeted LNA, which showed cell growth inhibitory effect in vitro and significant anti-tumor effects in an in vivo orthotopic bladder cancer model.

Compared to a subcutaneous transplant xenograft model, the orthotopic tumor implantation mimics the natural environment of bladder cancer, with intact pathological and immunological responses [18,19]. Most of the current methods for nucleic acid therapies are unsuitable for clinical use owing to their low uptake efficiency and high cytotoxicity [20]. Despite recent technological advances, achieving efficient nucleic acid drug delivery, particularly to extrahepatic tissues, remains a major translational limitation. In the case of bladder cancer, because a nucleic acid drug can be transurethrally administered, it is presumed that its pharmacological effect is expected to be expressed with lower side effects. Therefore, we believe that the miR-130 family targeted LNA is a promising nucleic acid drug for bladder cancer treatment.

Since miR-130b and miR-301a has been reported as oncomiR in colon cancer [12,13], we examined the effect of miR-130 family targeted LNA on colon cancer cell line HCT-116 cells and DLD-1 cells, which harbor constitutively active Src or not respectively [21]. Compared to DLD-1 cells, miR-130 family targeted LNA strongly suppressed cell proliferation of HCT-116 cells, suggesting that miR-130 family targeted LNA may have an anti-tumor effect not only on bladder cancer but also on colon cancer by targeting Src signaling pathway.

Overexpression of miR-130 family molecules upregulated the expression of phosphorylated Src (pSrc^Tyr416^), and miR-130 family targeted LNA downregulated pSrc^Tyr416^ in UM-UC-2 cells. Accumulating evidence indicates that PTPN1 is involved in cancer; however, conflicting findings suggest that it can exert both tumor-suppressing and tumor-promoting effects [22]. Our data showed that PTPN1 knockdown promoted tumor properties by upregulating the growth, invasion, and migration activities of UM-UC-2 cells. Moreover, PTPN1 knockdown upregulated pSrc^Tyr416^ expression in UM-UC-2 cells. Therefore, although the underlying mechanism should be addressed by in vitro and in vivo experiments, the miR-130 family may upregulate pSrc^Tyr416^ expression levels by downregulating PTPN1 in bladder cancer.

Our previous study showed that the miR-130 family upregulated the expression of phosphorylated FAK (pFAK^Tyr576^), which leads to increased migration and invasion in bladder cancer cells [8]. Src has been reported to activate FAK by phosphorylating FAK (pFAK^Tyr576^) [23], suggesting that the miR-130 family activate FAK by upregulating the expression of phosphorylated Src (pSrc^Tyr416^).

Integrins and RTKs are major upstream regulator for BCAR1 (p130CAS), mainly through the activation of Src and FAK, which phosphorylate BCAR1 and form a BCAR1-Src-FAK complex, leading to the activation of PI3K/Akt signaling pathway [24,25]. Our data showed that miR-130 family molecules upregulated the phosphorylation levels of BCAR1 (Table 1). Therefore, miR-130 family may promote migration and invasion activity via upregulating the expression of phosphorylated Src, leading to the activation of multiple tumor-promoting pathways in bladder cancer.

Jung et al. reported CDK2, CHEK1, and ERBB2 as central regulators mediating cisplatin resistance in bladder cancer [26]. Our phospho-proteomic data showed that overexpression of miR-130 family molecules upregulated the phosphorylation levels of CDK2, suggesting that miR-130 family might confer for cisplatin resistance in bladder cancer.

PTPN1 has been reported to suppress the PI3K/Akt signaling pathway by dephosphorylating the insulin receptor substrate (IRS) family [27]. Overexpression of the miR-130 family upregulated the expression of phosphorylated insulin-like growth factor 1 receptor (IGF1R) and insulin receptor (Table 1), which have been reported as PTPN1 substrates [28] in the IRS family. Moreover, it has been reported that IGF1R knockdown leads to the inhibition of cell growth in bladder cancer cells [29]. In addition, our previous report showed that the miR-130 family upregulated the expression of phosphorylated Akt (pAkt^Ser473^ and pAkt^Thr308^) by targeting PTEN expression in UM-UC-2 cells [8], suggesting that the miR-130 family upregulates the PI3K/Akt signaling pathway by both targeting PTPN1 and PTEN, leading to an aggressive phenotype in bladder cancer. The nucleic acid drug targeting the miR-130 family is expected to shed new light on bladder cancer treatment.

## 4. Materials and Methods

### 4.1. Reporter Plasmid Construction

To construct the reporter plasmids for miR-301a and PTPN1, the following oligonucleotide primers were used: hsa-miR-301a sense 5′-CTAGCGGCCGCTAGTGCTTTGACAATACTATTGCACTGG-3′, antisense 5′-TCGACCAGTGCAATAGTATTGTCAAAGCACTAGCGGCCGCTAGAGCT-3′; Sac I-PTPN1 3′-UTR sense 5′-GACTGAGCTCCATGCCGCGGTAGGTAAGG-3′, Sal I-PTPN1 antisense 5′-AACAGTCGACTACAACCGTCCTCCTTCCCA-3′. The annealed fragments or PCR products were digested with SacI and SalI and inserted into the pmirGLO dual-luciferase miRNA target expression vector (Promega, Madison, WI, USA). The 3′-UTRs of PTPN1 were cloned from the cDNA of 5637 cells using KOD-FX (Toyobo, Osaka, Japan).

### 4.2. Dual-Luciferase Reporter Assay

A pmirGLO dual-luciferase miRNA target expression vector was used for the 3′-UTR luciferase reporter assay (Promega). 5637 or UM-UC-2 cells were co-transfected with miR-130 family targeted LNA or miRNA mimic and reporter construct containing the predicted miR-130 family binding site in the 3′-UTR of the target genes. After 24 h of transfection, a dual-luciferase reporter assay was performed according to the manufacturer’s protocol (Promega). Luciferase activity was determined using a luminometer (Turner Biosystems 20/20 luminometer; Promega).

### 4.3. Cell Culture and Transfection

The human bladder cancer cell line 5637 was cultured in RPMI 1640 medium (Wako, Osaka, Japan) supplemented with 10% heat-inactivated fetal bovine serum (FBS) and 100 mg/L kanamycin at 37 °C under a 5% CO2 atmosphere. The miR-130 family targeted LNA and control LNA (5’-TCATACTA-3’) were synthesized by GeneDesign (Osaka, Japan). The designed sequences are as follows: LNA #1: ATTG5A5TT; LNA #2: ATTG5A5TG; LNA #3: A(L)^T(L)^5(L)^a^t^t^g^c^a^c^T(L)^G(L)^t; LNA #4: A(L)^T(L)^5(L)^a^t^t^g^c^a^5(L)^T(L)^g; LNA #5: A(L)^T(L)^5(L)^a^t^t^g^c^a^5(L)^T(L)^G(L); LNA #6: A(L)^t^5(L)^a^T(L)^t^G(L)^c^A(L)^c^T(L)^g; LNA #7: T(L)^A(L)^T(L)^c^a^t^t^g^c^a^5(L)^T(L)^g; LNA #8: A(L)^T(L)^T(L)^g^c^a^5(L)^T(L); LNA #9: A(L)^i^i^A(L)^T(L)^T(L)^G(L)^5(L)^A(L)^5(L)^T(L)^G; LNA #10: a^i^i^A(L)^T(L)^T(L)^G(L)^5(L)^A(L)^5(L)^T(L)^G(L); LNA #11: i^a^i^i^A(L)^T(L)^T(L)^G(L)^5(L)^A(L)^5(L)^T(L)^G(L). (L) and ^ indicate LNA and phosphorothioated binding, respectively. These LNAs were transfected using Lipofectamine 2000 reagent (Life Technologies, Carlsbad, CA, USA) following the manufacturer’s instructions. PTPN1 siRNAs (PTPN1 siRNA #1, Cat No. 4390824 and PTPN siRNA #2, Cat No. 4390824) and negative control siRNA (Silencer™ Select Negative Control No. 1 siRNA, Cat No. 4390843) were purchased from Sigma-Aldrich (St. Louis, MO, USA). For all siRNA transfection studies, 2 nmol/L siRNAs were transfected using Lipofectamine RNAiMAX reagent (Life Technologies).

### 4.4. Water-Soluble Tetrazolium Salt-8 (WST-8) Cell Growth Assay

Cell growth was examined using the WST-8 cell growth assay. The miR-130 family targeted LNA-transfected 5637 cells or UM-UC-3 cells were re-seeded in a 96-well plate (1 × 10^3^ cells/well and 0.5 × 10^3^ cells/well, respectively) 24 h after transfection and incubated for the indicated times. The PTPN1 siRNA-transfected UM-UC-2 cells were re-seeded in a 96-well plate (0.6 × 10^3^ cells/well) 24 h after transfection and incubated for the indicated times. After incubation for 3 h with WST-8 reagent (Dojindo, Osaka, Japan) at 37 °C, the optical density was recorded at a wavelength of 450/630 nm (measurement/reference) using a microplate reader (Bio-Rad, Hercules, CA, USA).

### 4.5. MiR-130-Targeted LNA Challenge on a Subcutaneous Xenograft Model

Eight female BALB/c Slc-nu/nu mice were obtained from Oriental Yeast (Tokyo, Japan). Five-week-old mice were used for the 5637 cell-xenograft experiments. Animals were kept under a 12 h light/dark cycle at 22–24 °C in a pathogen-free mouse facility. Food and water were provided ad libitum. Total of 5637 cells were adjusted to a concentration of 1.0 × 10^7^ cells suspended in 50 μL of serum-free RPMI 1640 medium. Cell suspensions with 50 μL of Μatrigel (Corning, Corning, NY, USA) were then injected subcutaneously into the right flanks of BALB/c nude mice (negative control, n = 8; miR-130F LNA, n = 8). Two weeks later, two randomly divided groups were treated with AteloGene^®^ Local Use “Quick Gelation” (KOKEN, Tokyo, Japan)-coated negative control LNA or miR-130F LNA (2 nmol/mouse). The tumor volume (V) was calculated as follows: V = (tumor length × tumor width^2^)/2. All procedures were performed according to a protocol approved by the Animal Experimentation Committee of Osaka University. The developed tumors were resected 32 days after the xenograft. Mice were euthanized by an overdose of isoflurane (two times the anesthetic dose) for 5 min.

### 4.6. Tumor Challenge on Orthotropic Bladder Cancer Model

An orthotopic animal model was established using a previously described technique [19]. Briefly, female mice were anesthetized with 1–2% isoflurane (Pfizer, New York, NY, USA). To prevent infection, the urethral tip was cleaned with 70% ethanol (Wako), and a 24-gauge Terumo catheter (Terumo) was inserted through the urethra into the bladder. The bladder was washed three times with 100 µL of phosphate-buffered saline (PBS; Thermo Fisher Scientific). Then, 100 µL of 0.25% trypsin (Life Technologies)/0.05% EDTA (Dojindo) at 37 °C was infused into the bladder under anesthesia, with each infusion retained for 5 min using a 30-g pressure clip (Natsume Seisakusho Co., Ltd., Tokyo, Japan, AM-1 30 g) on the urethra. The trypsin solution was then drained, and the bladder was washed twice with PBS (100 µL). Finally, 50 µL of DMEM containing 5 × 10^6^ UM-UC-3 Fluc cells was infused into the bladder and was retained for 2 h using a 30-g pressure clip under anesthesia. The clip was then removed, and the urethral meatus was disinfected with povidone-iodine. Cisplatin was intravenously injected into mice (10 mg/kg; PBS group, n = 5; cisplatin group, *n* = 5). LNA coated with AteloGene^®^ Local Use “Quick Gelation” (4 nmol/mouse) was injected by transurethral injection (negative control LNA group: N = 5, miR-130F LNA group: N = 5). Upon in vivo imaging, VivoGlo In Vivo Imaging Substrates (Promega) dissolved in PBS (10 mg/mL) were intraperitoneally injected. Then, the image was captured using NightOWL (Berthold Technologies; Calmbacher, Germany).

### 4.7. Construction of UM-UC-3 Cells Stably Expressing Luciferase Gene

The pGL4.51[luc2/CMV/Neo] vector (Promega) was transfected into UM-UC-3 cells and cultured in a medium containing 2 mg/mL G418 (Roche, Basel, Switzerland) for selection.

### 4.8. Proteome-Wide Tyrosine Phosphorylation Analysis

Proteome-wide tyrosine phosphorylation analysis was conducted using miRNA mimics of miR-301a, miR-301b, and miR-130b in UM-UC-2 cells. Proteome-wide tyrosine phosphorylation was assessed using a previously described technique [30]. Cell pellets were lysed in phase-transfer surfactant buffer supplemented with a complete protease inhibitor cocktail and PhosSTOP. Protein concentration was measured using a DC protein assay kit (Bio-Rad) according to the manufacturer’s protocol. Two milligrams of the protein lysate was reduced, alkylated, and subsequently trypsinized. Primary enrichment of phosphopeptides from 2 mg of the protein lysate was performed using Fe^3+^ IMAC resin. Phosphopeptides were labeled with the TMT 10-plex reagent according to the manufacturer’s protocol (Thermo Fisher Scientific, Waltham, MA, USA). A total of 20 mg of a labeled phosphopeptide mixture was prepared and then subjected to LC/MS analysis using a Q Exactive Plus mass spectrometer (Thermo Scientific, Waltham, MA, USA).

### 4.9. Western Blotting Analysis

Whole-cell lysates were separated by sodium dodecyl sulfate-polyacrylamide gel electrophoresis, and then transferred to a polyvinylidene difluoride (PVDF; Millipore, Billerica, MA, USA) membrane using a semidry transfer system (Bio-Rad). The membranes were probed with specific antibodies as indicated and then incubated with horseradish peroxidase-conjugated antibody against mouse or rabbit immunoglobulin (Cell Signaling Technology, Danvers, MA, USA), followed by the detection with enhanced chemiluminescence Western blotting detection reagents (GE Healthcare, Little Chalfont, UK). An ImageQuant LAS4000 mini system (GE Healthcare) was used to detect chemiluminescence. The following antibodies were used for immunological analysis in this study: anti-p-Src^Tyr416^ (Cell Signaling Technology, #2101), anti-Src (Sigma, SAB4300433), anti-Akt (Cell Signaling Technology, #4691), anti-p-Akt^Ser473^ (Cell Signaling Technology, #4060), anti-p-Akt^Thr308^ (Cell Signaling Technology, #13038), anti-PTPN1 (BD Transduction Laboratories, #610139), anti-p-EGFR^Tyr1068^ (Cell Signaling Technology, #3777), anti-EGFR (Cell Signaling Technology, #4267), and anti-β-actin (Sigma-Aldrich, A5316) antibodies.

### 4.10. SILAC-Based Proteome Analysis

For the SILAC experiments, Dulbecco’s modified Eagle’s medium without l-arginine and l-lysine (Invitrogen, Carlsbad, CA, USA) was supplemented with dialyzed FBS (Invitrogen). The medium was then divided into three portions and supplemented with ^13^C_6_,^15^ N_4_
l-arginine, ^13^C_6_, ^15^N_2_
l-lysine, ^13^C_6_
l-arginine, and 4,4,5,5-D_4_
l-lysine or normal l-arginine and l-lysine, to produce “heavy” or “medium” or “light” SILAC medium, respectively. All isotope-labeled l-arginine and l-lysine were purchased from Silantes (Gollierstraße, München, Germany). UM-UC-2 cells were grown in SILAC medium for two weeks to ensure that the amino acids had been fully incorporated. Protein lysates were examined by LC/MS analysis using a Q Exactive Plus mass spectrometer (Thermo Scientific) at the National Institutes of Biomedical Innovation, Health and Nutrition.

### 4.11. Cell Invasion Assay

The BioCoat tumor invasion system (Corning) was used to perform the cell invasion assay. UM-UC-2 cells transfected with PTPN1 siRNA or negative control siRNA were seeded in a 96-well plate (3 × 10^3^ cells/well). Following incubation for 22 h, the cells were labeled with calcein-AM (2 μg/mL, Takara-Bio, Shiga, Japan), and the fluorescence of the invading cells was recorded at a wavelength of 494/517 nm (Ex/Em).

### 4.12. Wound Healing Assay

UM-UC-2 cells were transfected with PTPN1 siRNA 24 h after seeding in a 12-well plate (6 × 10^4^ cells/well) and incubated for 72 h. A wound was created in the monolayer of UM-UC-2 cells at ~90% confluence using a sterile 1-mL pipette tip. Cell images were recorded at 0 and 24 h after wound creation using a fluorescence microscope (Olympus, Tokyo, Japan). Cell migration activity was calculated as follows: (wound area 24 h after wound creation)–(wound area 0 h after wound creation).

### 4.13. Statistics

The results were expressed as the mean ± standard deviation (SD). Differences between values were statistically analyzed using Student’s t-test or one-way analysis of variance (ANOVA) with Bonferroni post-hoc tests (GraphPad Prism 5.0, GraphPad Software, La Jolla, CA, USA). Statistical significance was set at *p* < 0.05. Normal distribution and homoscedasticity were tested by Shapiro–Wilk test and Lèvene’s test respectively (XLSTAT software, Paris, France).

## Figures and Tables

**Figure 1 ijms-22-04751-f001:**
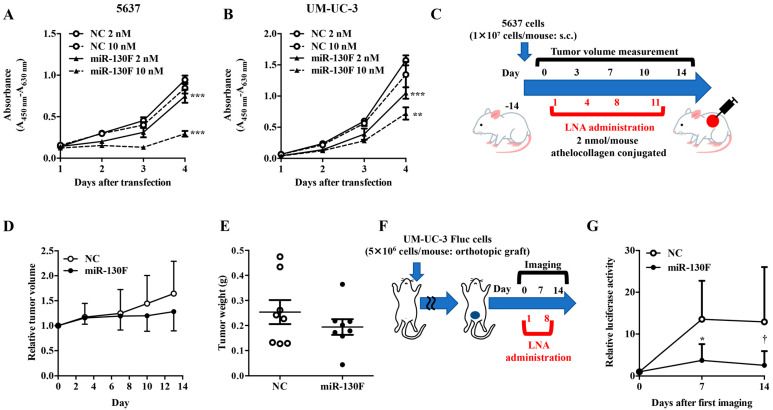
Functional verification of miR-130 family targeted LNAs in bladder cancer cells. The effect of miR-130 family targeted LNA (miR-130F) on the proliferation of 5637 (**A**) or UM-UC-3 (**B**) cells was measured using a WST-8 assay. Data are presented as the mean ± SD (*n* = 4). Data were analyzed using one-way ANOVA with Bonferroni post-hoc tests ** *p* < 0.01, *** *p* < 0.001. (**C**) Evaluation scheme of miR-130 family targeted LNA (miR-130F) in a 5637 cell-xenograft model. Control LNA (NC) and miR-130F conjugated with athelocollagen were administered at a dose of 2 nmol/mouse into 5637 cell-xenograft mice. (**D**) Relative tumor volume was calculated using the formula: tumor volume [mm^3^] = (major axis [mm]) × (minor axis [mm])^2^ × 0.5. Tumors resected on day 14 were weighed (**E**). Data are presented as the mean ± SD (*n* = 8). (**F**) Evaluation scheme of miR-130 family targeted LNA (miR-130F) in an orthotopic bladder cancer model. (**G**) Control LNA (NC) and miR-130F were transurethrally administered at a dose of 4 nmol/mouse in UM-UC-3 cell-inoculated mice. Relative tumor volume was measured by in vivo imaging. Data are presented as the mean ± SD (*n* = 5). Data were analyzed using the Mann–Whitney U test * *p* < 0.05; † *p* = 0.05.

**Figure 2 ijms-22-04751-f002:**
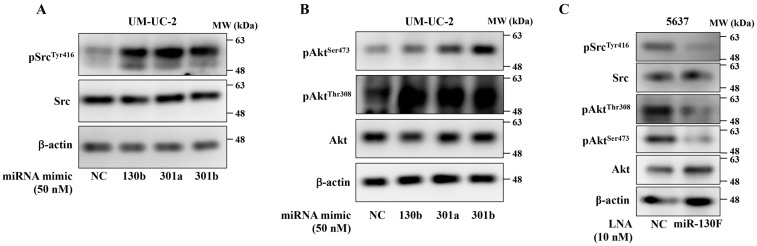
miR-130 family functions as oncomiR by upregulating Src phosphorylation in bladder cancer cells. miR-130 family mimics were transfected into UM-UC-2 cells. Protein expression levels of pSrc^Tyr416^, Src, pAkt^Ser473^ (**A**), pAkt^Thr308^, Akt (**B**), and β-actin were evaluated by Western blot analysis. Representative pictures of three independent experiments are shown. Uncropped Western blot data are shown in Appendix A. (**C**) 5637 cells were transfected with miR-130F. Protein expression levels of pSrc^Tyr416^, Src, pAkt^Ser473^, pAkt^Thr308^, Akt, and β-actin were evaluated by Western blot analysis. Representative pictures of three independent experiments are shown. Uncropped Western blot data are shown in Appendix A.

**Figure 3 ijms-22-04751-f003:**
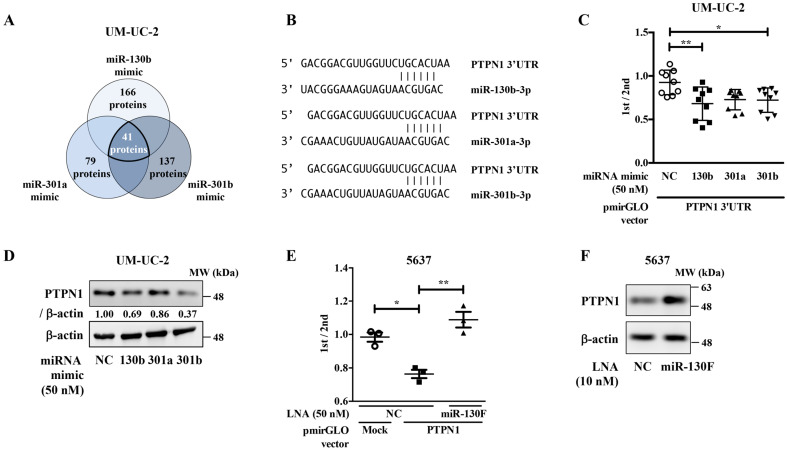
The miR-130 family target PTPN1 in bladder cancer. (**A**) SILAC-based proteome analysis was conducted using negative control, miR-301a, miR-301b, or miR-130b mimic-transfected UM-UC-2 cells. A total of 41 proteins were shared among the three members of the miR-130 family. (**B**) Predicted miR-130 family binding sequences within the 3′-UTR of human PTPN1 gene. The number indicates the nucleotide position of the predicted miR-130 family binding site from the start of the PTPN1 3′-UTR. (**C**) A dual-luciferase reporter assay was performed in UM-UC-2 cells transfected with negative control (NC) or miR-130 family mimics. Data are presented as the mean ± SD of five independent experiments. Data were analyzed using one-way ANOVA with Bonferroni post-hoc tests * *p* < 0.05; ** *p* < 0.01. Data are presented as the mean ± SD (*n* = 9). (**D**) UM-UC-2 cells were transfected with miR-130 family mimics. Protein expression levels of PTPN1 and β-actin were evaluated by Western blot analysis. Representative pictures of three independent experiments are shown. The numbers in the figure indicate the expression ratio of PTPN1 and actin analyzed by the densitometer. Uncropped Western blot data are shown in Appendix A. (**E**) A dual-luciferase reporter assay was performed in 5637 cells transfected with negative control (NC) or miR-130F. Data were analyzed using one-way ANOVA with Bonferroni post-hoc tests * *p* < 0.05; ** *p* < 0.01. Data are presented as the mean ± SD (*n* = 3). (**F**) 5637 cells were transfected with miR-130F. Protein expression levels of PTPN1 and β-actin were evaluated by Western blot analysis. Representative pictures of three independent experiments are shown. Uncropped Western blot data are shown in Appendix A.

**Figure 4 ijms-22-04751-f004:**
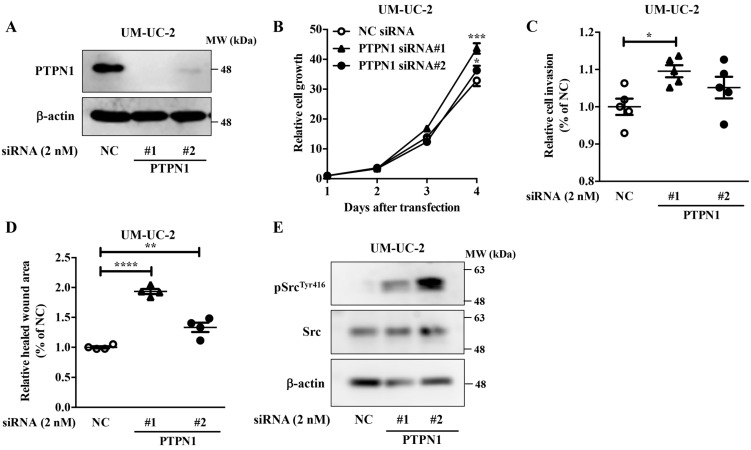
PTPN1 function as a tumor suppressor in bladder cancer cells. PTPN1 functions as a tumor suppressor in bladder cancer cells. UM-UC-2 cells were transfected with negative control siRNA (NC) or PTPN1 siRNAs. (**A**) Protein expression levels of PTPN1 and β-actin were evaluated by Western blot analysis. Representative pictures of three independent experiments are shown. Uncropped Western blot data are shown in Appendix A. The effects of PTPN1 siRNAs on the growth (**B**), invasion (**C**), and migration (**D**) of UM-UC-2 cells were examined. Data are presented as the mean ± SD (Figure 4B: *n* = 3, Figure 4C: *n* = 5, Figure 4D: *n* = 4). Data were analyzed using one-way ANOVA with Bonferroni post-hoc tests * *p* < 0.05; ** *p* < 0.01; *** *p* < 0.001; **** *p* < 0.0001. (**E**) Western blot analyses of pSrc^Tyr416^, Src, and β-actin were performed using PTPN1 siRNA-transfected UM-UC-2 cells. Representative pictures of three independent experiments are shown. Uncropped Western blot data are shown in Appendix A.

**Table 1 ijms-22-04751-t001:** Tyrosine-phosphorylated proteins upregulated by miR-130 family mimic.

Protein Name	Position	miR-130b	miR-301a	miR-301b	Localization Probability	Posterior Error Probability	Score
ABI1	213	1.66	1.57	1.50	0.996	3.E-12	69.361
BCAR1	128	1.72	1.78	1.82	1.000	8.E-13	76.301
CDK1	15	1.90	1.59	1.56	1.000	1.E-29	145.88
CDK16; CDK17	176; 203	1.88	1.43	1.43	0.986	4.E-06	56.043
CDK2; CDK3	15; 15	1.50	1.23	1.30	1.000	1.E-14	122.64
CLASP2	1022	4.19	3.67	2.05	0.890	1.E-06	54.281
DDX5; DDX17	202; 279	1.81	1.78	1.28	1.000	2.E-04	74.841
EPHA2	575	1.80	1.76	1.53	0.999	2.E-05	89.301
EPHA2	588	2.98	2.49	2.23	0.850	4.E-36	130.83
EPHA2	594	2.98	2.45	2.23	0.800	5.E-42	155.37
EPHA2	772	2.45	1.93	1.82	1.000	3.E-47	164.88
FYN; YES1; SRC; LCK	420; 426; 419; 394	2.29	1.64	1.68	0.996	1.E-25	146.16
GART	348	2.70	3.28	2.60	0.979	2.E-03	54.157
HIST1H2BL; HIST1H2BM; HIST1H2BN; HIST1H2BH; HIST2H2BF; HIST1H2BC; HIST1H2BD; H2BFS; HIST1H2BK	41; 41; 41; 41; 41; 41; 41; 41; 41	1.91	1.64	1.93	0.999	3.E-05	87.476
HIST3H2BB; HIST2H2BE; HIST1H2BB; HIST1H2BO; HIST1H2BJ; HIST2H2BD; HIST2H2BC	41; 41; 41; 41; 41; 41; 41	2.73	2.78	3.02	0.999	3.E-05	87.476
HSPA9	118	1.89	1.30	1.73	0.991	2.E-18	115.1
IGF1R; INSR	1165; 1189	2.02	1.81	1.58	0.881	1.E-04	79.148
INPPL1	1162	2.27	1.63	1.69	0.995	2.E-24	125.73
LDHA	239	3.85	2.83	3.47	1.000	2.E-07	94.767
LYN	306	1.82	1.50	1.69	1.000	8.E-05	94.662
MAPK14	182	1.91	1.53	2.41	0.969	2.E-10	96.756
MKI67	1552	2.04	2.04	2.03	0.998	3.E-03	44.543
MRPL22	165	1.60	1.34	1.62	1.000	3.E-03	46.426
PPP1CA	306	1.70	1.42	1.39	1.000	3.E-19	117.53
PRKCD	313	1.90	2.05	1.46	1.000	5.E-33	139.74
PRPF4B	849	2.38	1.98	2.71	0.954	6.E-20	98.543
RAB2B	3	2.01	1.49	1.60	0.856	2.E-02	67.032
RHOT1	465	2.54	3.58	4.51	0.824	2.E-08	78.69
SHROOM1	33	1.65	1.50	1.44	0.861	3.E-06	51.495
SSBP1	101	1.94	1.55	1.79	0.998	1.E-06	95.502
TJP2	1118	1.56	1.67	1.76	1.000	2.E-04	59.908
TYRO3; MERTK	685; 753	1.88	1.35	1.36	0.992	6.E-08	108.74

**Table 2 ijms-22-04751-t002:** Candidate target proteins of miR-130 family.

Protein Name	
HMGN4	RPL24
ARPC2	SRSF3
HNRNPR	SRSF2
BUB3	CAP1
WDR1	SLC7A5
HIST1H1E	TWF1
HSPA8	HNRNPA0
PTPN1	EIF3I
VCL	PTGES3
MAP4	MAPRE1
MAPK1	TIMM50
RPL13A	PABPN1
LRPPRC	SERBP1
RPL5	FERMT2
METAP1	FUBP1
ADK	ERP44
MTPN	SFXN1
ABCE1	TMOD3
RPS15A	SHPK
RPL8	CDV3
SRPRB	

## Data Availability

The data that support the findings of this study are available from the corresponding author, upon reasonable request.

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
