# Peer review of "Pharmacological Inhibition of miR-130 Family Suppresses Bladder Tumor Growth by Targeting Various Oncogenic Pathways via PTPN1"

_ijms, 2021, doi:10.3390/ijms22094751_

Round 1

Reviewer 1 Report

Several minor points that need attention: 

1st:  The authors should also validate their findings of RTKs in their cell lines (change in phosphorylation of EGFR/VEGFR/HER2/etc in bladder cell lines).

2nd:  The authors should consider validating their in vitro pathway analysis and conjecture on PTEN/SRC with tumor samples form their in vivo experiments.  This would strengthen their assertion to the possible mechanism of MiR-130’s action in vivo.

3rd:  The authors don’t use any cell lines other than bladder cancer to show portability (or not) to other cancer models.  Perhaps using models of EGFR (A431) WT would show a similar correlation.

4th:  The discussion does not adequately extrapolate the link between the genetics of bladder cancer and the impact of PTEN.  In general, bladder cancer is not driven by RTK signaling, so making the link between the phospho-proteomic data and common cisplatin targets might be worthwhile. 

Suggested corrections to Figures/Tables:

Figure 2A.  This figure panel needs clarification.  If this data was obtained from the phospho-proteomics experiments then the X-axis should be proteins not genes.  Furthermore, many of the proteins listed function using the same pathway (EGFR and HER2 form hetero-dimers) thus the Y axis is incorrect.  This should be changed to protein instead of signaling pathway.

Table 1:

Please include values for confidence scores for each phosphorylation site.  Authors used Thermo instruments so they should have access via Proteome Discoverer software by the instrument manufacturer.

Table 2:

Authors need to supply data pertaining to # peptides; # unique peptides; # protein hits; protein confidence score; SILAC ratio with error must be given.

Figure 3A.  The figure indicates gene(s); This data is derived from protein IDs and the figure should be changed to reflect this.

Author Response

We have revised the manuscript according to the editorial board members’ and reviewers’ helpful suggestions (indicated in bold type) and have included additional information (indicated in red in the revised manuscript). In the revised manuscript, we changed the order of co-authors who conducted additional experiments (Yuya Monoe for Supplementary Fig. 2, Supplementary Fig. 5). The revised manuscript consists of 4 Figures, 6 Supplementary Figures, and 2 Supplementary Tables.

Reviewer #1

The authors should also validate their findings of RTKs in their cell lines (change in phosphorylation of EGFR/VEGFR/HER2/etc in bladder cell lines).

We thank the reviewer for this constructive suggestion. Although we have performed western blot analysis using p-EGFRTyr1068 antibody, the miR-130 family mimics had no significant effect on phosphorylation levels of p-EGFRTyr1068 (Supplementary Fig. 5 in revised manuscript). Therefore, we think that miR-130 family functions as an oncomiR not by directly targeting RTK but by targeting various tumor-promoting pathway through Src phosphorylation. We have added the result and revised the manuscript (page 3, line 36; page 8, lines 126-128; page 8, lines 130-134; page 18, lines 319-320).

The authors should consider validating their in vitro pathway analysis and conjecture on PTEN/SRC with tumor samples form their in vivo experiments.  This would strengthen their assertion to the possible mechanism of MiR-130’s action in vivo.

We thank the reviewer for this constructive suggestion. FFPE samples were prepared from bladder tissue in Fig. 1G and HE staining were conducted. Although NC LNA-treated bladder tissue showed clear tumor image, we could not recognize the tumor image in miR-130F LNA-treated bladder tissue (Figure). We would also be interested in the effect miR-130F LNA on PTEN/SRC in vivo. We have now acknowledged this and suggested it as a topic for further research in the Discussion section of the revised manuscript (page 11, line 191).

Figure. HE-staining image of LNA-treated mouse bladder tissue

The authors don’t use any cell lines other than bladder cancer to show portability (or not) to other cancer models.  Perhaps using models of EGFR (A431) WT would show a similar correlation.

We strongly appreciate the reviewer's comment on this point. We have conducted cell proliferation assay using HCT-116 cells and DLD-1 cells, which harbors constitutively active Src or not respectively. The miR-130F LNA successfully suppressed the proliferation of HCT-116 cells and DLD-1 cells. Moreover, the effect of miR-130F LNA was stronger in HCT-116 cells compared to DLD-1 cells. These results suggests that miR-130F LNA may have anti-tumor potential not only for bladder cancer but also for colon cancer via targeting Src signaling pathway (Supplementary Figs. 2A and B, revised manuscript: page 6, lines 95-98; page 11, lines 178-183).

The discussion does not adequately extrapolate the link between the genetics of bladder cancer and the impact of PTEN.  In general, bladder cancer is not driven by RTK signaling, so making the link between the phospho-proteomic data and common cisplatin targets might be worthwhile. 

We thank the reviewer for this constructive suggestion. According to the reviewer’s helpful suggestions, we have found the link between phosphor-proteomic data and cisplatin target and incorporated in Discussion section of the revised manuscript (page 11, lines 193-196).

Figure 2A. This figure panel needs clarification. If this data was obtained from the phospho-proteomics experiments then the X-axis should be proteins not genes. Furthermore, many of the proteins listed function using the same pathway (EGFR and HER2 form hetero-dimers) thus the Y axis is incorrect. This should be changed to protein instead of signaling pathway.

 According to the reviewer’s helpful suggestions, we have incorporated the result of gene enrichment analysis using WikiPathways (https://www.wikipathways.org/index.php/WikiPathways), which enable more detailed classification and revised to supplementary table 1 (revised manuscript: page 7, line 120 to page 8, line 122).

Table 1: Please include values for confidence scores for each phosphorylation site. Authors used Thermo instruments so they should have access via Proteome Discoverer software by the instrument manufacturer.

Thank you for pointing out our insufficiency. We added the values of confidence scores (Localization probability, Posterior error probability, Score) for each phosphorylation site in revised table 1.

Table 2: Authors need to supply data pertaining to # peptides; # unique peptides; # protein hits; protein confidence score; SILAC ratio with error must be given.

Thank you for pointing out our insufficiency. We added the information of # peptides; # unique peptides; # protein hits; protein confidence score (Posterior error probability, Score); SILAC ratio with error (Ratio H/L variability) and revised to supplementary table 2 (revised manuscript: page 9, lines 142-143).

Figure 3A. The figure indicates gene(s); This data is derived from protein IDs and the figure should be changed to reflect this.

Thank you for pointing out our insufficiency. We have amended the description in Fig. 3A.

Reviewer 2 Report

Dear authors,

This work analyses the effect of pharmacological inhibition of miR-130 family in bladder tumour growth and the results show a relevant information about the possible use of miR-130 family-targeting LNA as therapeutic agent for bladder cancer. The manuscript is well written and structured, the design is appropriate, and the results are clearly presented. However, some minor revisions are necessary. First, the latin names (“in vivo” and “in vitro”) should be written in italics, and second, before conducting a parametric analysis such as Student's t-test or ANOVA, the authors should ensure that their data follow a normal distribution and show homoscedasticity. A normality test (for example, Shapiro-Wilk) and a homoscedasticity test (for example, Lèvene) should be performed.

I hope these recommendations improve the quality of manuscript.

Author Response

We have revised the manuscript according to the editorial board members’ and reviewers’ helpful suggestions (indicated in bold type) and have included additional information (indicated in red in the revised manuscript). In the revised manuscript, we changed the order of co-authors who conducted additional experiments (Yuya Monoe for Supplementary Fig. 2, Supplementary Fig. 5). The revised manuscript consists of 4 Figures, 6 Supplementary Figures, and 2 Supplementary Tables.

Reviewer #2

This work analyses the effect of pharmacological inhibition of miR-130 family in bladder tumour growth and the results show a relevant information about the possible use of miR-130 family-targeting LNA as therapeutic agent for bladder cancer. The manuscript is well written and structured, the design is appropriate, and the results are clearly presented. However, some minor revisions are necessary. First, the latin names (“in vivo” and “in vitro”) should be written in italics, >> [Ans. 1-1]

and second, before conducting a parametric analysis such as Student's t-test or ANOVA, the authors should ensure that their data follow a normal distribution and show homoscedasticity. A normality test (for example, Shapiro-Wilk) and a homoscedasticity test (for example, Lèvene) should be performed. I hope these recommendations improve the quality of manuscript.                                                                                        >> [Ans. 1-2]

[Ans. 1-1] We thank the reviewer for this constructive suggestion. We have amended the description in the manuscript (revised manuscript: page 3, line 34; page 6, lines 91-92; page 6, line 95; page 6, line 99; page 7, line 107; page 10, lines 166-167; page 11, line 191; page 16, line 287).

[Ans. 1-2] We appreciate the editorial board members’ concerns regarding this point. We have checked the normal distribution (Shapiro-Wilk test) and homoscedasticity (Lèvene’s test). We have added the description in the manuscript (revised manuscript: page 20, lines 350-351).